# Blood Pressure Patterns in Patients with Parkinson’s Disease: A Systematic Review

**DOI:** 10.3390/jpm11020129

**Published:** 2021-02-15

**Authors:** Delia Tulbă, Liviu Cozma, Paul Bălănescu, Adrian Buzea, Cristian Băicuș, Bogdan Ovidiu Popescu

**Affiliations:** 1Department of Neurology, Colentina Clinical Hospital, 20125 Bucharest, Romania; delia.tulba@umfcd.ro (D.T.); liviu.cozma@drd.umfcd.ro (L.C.); 2Colentina–Research and Development Center, Colentina Clinical Hospital, 20125 Bucharest, Romania; paul.balanescu@umfcd.ro (P.B.); cristian.baicus@umfcd.ro (C.B.); 3“Carol Davila” University of Medicine and Pharmacy, 020021 Bucharest, Romania; catalin.buzea@umfcd.ro; 4Department of Cardiology, Colentina Clinical Hospital, 20125 Bucharest, Romania; 5Department of Internal Medicine, Colentina Clinical Hospital, 20125 Bucharest, Romania; 6Laboratory of Cell Biology, Neurosciences and Experimental Myology, “Victor Babeș” National Institute of Pathology, 050096 Bucharest, Romania

**Keywords:** Parkinson’s disease, 24-h ambulatory blood pressure monitoring, blood pressure patterns, arterial hypertension, dysautonomia, orthostatic hypotension, supine hypertension, nocturnal hypertension, reduced dipping, reverse dipping

## Abstract

(1) Background: Cardiovascular autonomic dysfunction is a non-motor feature in Parkinson’s disease with negative impact on functionality and life expectancy, prompting early detection and proper management. We aimed to describe the blood pressure patterns reported in patients with Parkinson’s disease, as measured by 24-h ambulatory blood pressure monitoring. (2) Methods: We conducted a systematic search on the PubMed database. Studies enrolling patients with Parkinson’s disease undergoing 24-h ambulatory blood pressure monitoring were included. Data regarding study population, Parkinson’s disease course, vasoactive drugs, blood pressure profiles, and measurements were recorded. (3) Results: The search identified 172 studies. Forty studies eventually fulfilled the inclusion criteria, with 3090 patients enrolled. Abnormal blood pressure profiles were commonly encountered: high blood pressure in 38.13% of patients (938/2460), orthostatic hypotension in 38.68% (941/2433), supine hypertension in 27.76% (445/1603) and nocturnal hypertension in 38.91% (737/1894). Dipping status was also altered often, 40.46% of patients (477/1179) being reverse dippers and 35.67% (310/869) reduced dippers. All these patterns were correlated with negative clinical and imaging outcomes. (4) Conclusion: Patients with Parkinson’s disease have significantly altered blood pressure patterns that carry a negative prognosis. Ambulatory blood pressure monitoring should be validated as a biomarker of PD-associated cardiovascular dysautonomia and a tool for assisting therapeutic interventions.

## 1. Introduction

### 1.1. Background/Rationale

Parkinson’s disease (PD) is the second most common neurodegenerative disorder, with a prevalence of 0.3% of the entire population in industrialized countries, reaching up to 4% in the highest age groups [1]. The biomedical and economic burden of PD is becoming more and more evident with population ageing [1,2]. Cardiovascular autonomic dysfunction manifested as abnormal arterial blood pressure (BP) patterns is part of the spectrum of non-motor features of PD that occur across all stages (including prodromal) and contribute to impaired quality of life, disability and shorter life expectancy [3], therefore requiring proper assessment and management.

Considering the high prevalence of high blood pressure (HBP) and cardiovascular disorders (CVD) in the elderly, these are likely comorbidities in PD patients, warranting a closer look at their interrelation. Although HBP does not seem to increase the risk of PD [4], it might increase the mortality in this group of patients through several mechanisms. Apart from contributing to CVD [5] which is the most commonly reported cause of death in PD patients [6], HBP complicates dysautonomic manifestations in PD such as orthostatic hypotension (OH) or postprandial hypotension. HBP treatment worsens OH which is responsible for falls, physical deconditioning, cognitive decline and cardiovascular events [7], whereas OH treatment might aggravate HBP, possibly resulting in end-organ damage. There are currently no guideline recommendations regarding the follow-up and therapeutic management of coexistent HBP and OH.

In PD patients with autonomic dysfunction and/or HBP, BP measurement might lead to wrong assumptions unless properly performed. Mistaking supine hypertension (SH) for HBP (or vice versa) or missing HBP in case of coexistent OH are just two clinical scenarios that could lead to wrong treatment decisions. With PD symptoms fluctuating throughout the day, it is hard to believe that a conventional office BP measurement reflects the real BP profile of the patient, especially if white-coat effect and white-coat hypertension are superimposed. Therefore, out-of-office BP measurements such as home BP monitoring and ambulatory BP monitoring (ABPM) might be more suitable. Apart from being a better predictor of hypertension-mediated organ damage and cardiovascular outcomes than conventional office BP measurement, 24-h ABPM (especially when coupled with a diary of the patient’s activities and sleep time) provides useful information about nocturnal BP profiles and activity-related BP [8].

### 1.2. Objectives

We aimed to describe the BP patterns reported in patients with PD, as measured by 24-h ABPM, in conjunction with HBP and dysautonomia. We focused on 24-h, daytime and nighttime mean values of systolic BP (SBP) and diastolic BP (DBP), BP variability, dipping status, occurrence of OH and SH as well as HBP diagnosis, in an attempt to describe the circadian rhythm and variability of BP in PD. We hypothesized that these patients have abnormal BP patterns mainly due to autonomic impairment and/or superimposed HBP.

## 2. Materials and Methods

### 2.1. Protocol

The protocol for this systematic review was conceived a priori based on PRISMA 2009 Checklist.

Study eligibility and search. We designed a systematic review based on the following research question:

“What are the blood pressure patterns identified by 24-h ambulatory blood pressure monitoring in patients with Parkinson’s disease?”

The target population consisted of adult patients diagnosed with PD. The intervention was ABPM performed during a minimum of 24 h. The outcomes were BP profiles (HBP, OH, SH, nocturnal hypertension (NH), dipping status) and BP measurements (mean BP during daytime, nighttime as well as 24-h, median variation coefficient and BP load).

We conducted a systematic search on the PubMed database on the 31st of July 2020, scrutinizing the studies that enrolled patients with PD undergoing 24-h ABPM. We performed the search again on 2nd of December 2020 to identify any recently published studies meeting the inclusion criteria. Articles were included from inception. No restriction was applied, and the following search strategy was used:

((((((((((((“blood pressure”[MeSH Terms] OR (“blood”[All Fields] AND “pressure”[All Fields])) OR “blood pressure”[All Fields]) OR “blood pressure determination”[MeSH Terms]) OR ((“blood”[All Fields] AND “pressure”[All Fields]) AND “determination”[All Fields])) OR “blood pressure determination”[All Fields]) OR (“blood”[All Fields] AND “pressure”[All Fields])) OR “blood pressure”[All Fields]) OR “arterial pressure”[MeSH Terms]) OR (“arterial”[All Fields] AND “pressure”[All Fields])) OR “arterial pressure”[All Fields]) OR (“blood”[All Fields] AND “pressure”[All Fields])) AND (((((((((((((“monitors”[All Fields] OR “monitorable”[All Fields]) OR “monitored”[All Fields]) OR “monitoring”[All Fields]) OR “monitorings”[All Fields]) OR “monitoring, physiologic”[MeSH Terms]) OR (“monitoring”[All Fields] AND “physiologic”[All Fields])) OR “physiologic monitoring”[All Fields]) OR “monitor”[All Fields]) OR “monitorings”[All Fields]) OR “monitorization”[All Fields]) OR “monitorize”[All Fields]) OR “monitorized”[All Fields]) OR “monitors”[All Fields])) AND (“Parkinson’s disease”[All Fields] OR “Parkinson Disease”[MeSH Terms])

Studies that were not identified through this search strategy but were subsequently identified in the review studies were also considered for inclusion.

Inclusion criteria were: any study (either descriptive or analytic) enrolling adult patients diagnosed with PD (either by UK Parkinson’s Disease Society Brain Bank Diagnostic Criteria or Movement Disorder Society Clinical Diagnostic Criteria for Parkinson’s disease) who underwent an ABPM performed during a minimum of 24-h. Exclusion criteria were: studies with unavailable data on 24-h ABPM measurements or unsuitable reports (e.g., graphs that could not provide the exact mean value of BP), studies addressing invasive blood pressure monitoring or the cardiovascular effects of a recently prescribed (possible vasoactive) drug (unless they included data on prior BP profile), unavailability of full-text, other languages than English.

### 2.2. Study Appraisal

Two authors (D.T. and L.C.) independently evaluated the studies and included the eligible ones, as stated above. Differences were discussed with the other authors until reaching a consensus. The first evaluation covered the abstracts, then the full-text articles were assessed whenever possible to select the relevant ones. Duplicates were excluded.

Data regarding study population (number of PD patients, sex, mean age), PD progression and staging (PD duration, Hoehn and Yahr stage, Unified Parkinson’s Disease Rating Scale (UPDRS) score), potentially vasoactive drugs (e.g., antihypertensives, antihypotensives, antiparkinsonians, antidepressants, antipsychotics, etc.), BP profiles (HBP, OH, SH, NH, dipping profile) and measurements (24-h, diurnal and nocturnal mean SBP/DBP, BP median coefficient variation and BP load) were extracted. We took into consideration both prior and newly diagnosed OH (either by bedside assessment, head-up Tilt test or ABPM + diary) and HBP (either by office BP measurement or ABPM), whereas 24-h ABPM was mandatory for BP measurements. For classifying BP profiles, we used the definitions provided by the guidelines and/or consensus statements, as follows:

HBP: “office SBP values ≥ 140 mmHg and/or DBP values ≥ 90 mmHg” or ≥130/80 mmHg on 24-h ABPM [8]

OH: “reduction in SBP of ≥20 mmHg or in DBP of ≥10 mmHg within 3 min of standing” [8]

SH: “In patients with proven OH, SH is defined as systolic BP of ≥140 mmHg and/or diastolic BP of ≥90 mmHg, measured after at least 5 min of rest in the supine position.” [9]

NH: “nighttime BP ≥ 120/70 mmHg” [10]

Dipping: “nocturnal BP fall ≥10% of the daytime average BP value” [8]. It includes both normal dipping (≥10% and <20%) and extreme dipping (≥20%). Reduced dipping: “mean nocturnal BP reduction of <10% with respect to mean daytime BP values” [9]. Reverse dipping: “when the mean BP does not decrease or even increases during the night with respect to daytime” [9]. Although not unanimously accepted, we considered that non-dipping comprises both reduced and reverse dipping since most studies employed this definition. However, a Position Paper on ABPM from the European Society of Hypertension uses other dipping categories, namely extreme dipping, dipping (for normal dipping), mild dipping (for reduced dipping) and rising or absence of dipping (for reverse dipping) [11].

For variables reported as mean values (such as BP measurements) in several subgroups of the PD population, we calculated and recorded the weighted average. We also reported the total number of PD patients with HBP, OH, SH, NH and reduced/reverse dipping as well as their percentage out of the total number of patients evaluated in this regard; we emphasize the fact that these data were not available for all the studies. BP variability was evaluated by the coefficient of variation. Unless it was reported, we calculated the median 24-h, daytime and nighttime coefficient of variation for both SBP and DBP by dividing the reported standard deviation (SD) by the average BP level and multiplying by 100 whenever it was possible.

## 3. Results

The search on PubMed database identified 172 results, with one duplicate removed. Two additional studies were identified in a review study and were subsequently included. After screening by title and abstract, 88 studies were excluded. Those accepted were consequently read full-text and 40 studies fulfilled the inclusion criteria, as illustrated in the flow chart (Figure 1). All the studies included were cross-sectional, most of them being prospective.

Demographics and PD status are detailed in Table 1. A total number of 3090 PD patients were included in the selected studies. Considering that two studies [12,13] did not mention the sex of the patients, there were 1550 males (52.04%) and 1431 females (47.95%), with a sex ratio of 1.08. The mean age of the patients was >60 years in all the studies, with one study only reporting the median age of the population [14]; none of studies specified the number of patients with young, middle-aged, and late-onset PD. The mean duration of PD (i.e., from the onset of motor symptoms) ranged from 1 to 18.4 years, with 7 studies not addressing this issue or reporting only the median duration. The mean value of Hoehn and Yahr scale ranged from 1.5 to 3.6 (missing in 16 studies), while mean motor UPDRS ranged from 14.5 to 36.2 (missing in 25 studies). Various studies enrolled control groups (healthy control subjects and/or patients with HBP, pure autonomic failure, Lewy body dementia, multiple system atrophy, progressive supranuclear palsy, drug-induced parkinsonism, SWEDDs—scans without evidence of dopaminergic deficit or mild cognitive decline) or performed an analysis of PD subgroups (i.e., with and without autonomic failure, according to dipper and non-dipper profile, with or without depression, according to the duration of the disease).

Intake of vasoactive drugs is listed in Appendix A. Considerable differences in reporting it were noticed. For instance, when referring to antiparkinsonian drugs, some studies reported the levodopa equivalent dose while others rather mentioned the drug class or the specific agent, the number of patients taking the medication or the dosage. Moreover, some studies did not even record the intake of vasoactive drugs or specifically withheld them during ABPM, with possible implications for BP profiles and measurements. Interestingly, in one study dopaminergic therapy did not seem to increase the occurrence of OH, SH, NH and non-dipping [33], whereas others did not analyze this association.

BP profiles are recorded in Table 2 and Appendix A, encompassing HBP, OH, SH, NH, and dipping status, all with different significance and prognosis (see Discussion). However, only a few studies with inconsistent results performed a subanalysis of BP profiles in terms of outcomes, therefore the correlations reported virtually apply to a small number of patients and require further checking.

We identified 938 patients with HBP within PD group (38.13% of the number of patients who were evaluated for this issue) in 26 studies, whereas 5 studies designated HBP as exclusion criterion for patient enrollment and 9 studies did not address this issue. It is curious that no subanalysis of PD patients with HBP was performed, therefore there is no available data regarding the interaction between HBP and OH in PD patients. In two studies, PD patients with OH had a slightly lower prevalence of HBP compared to non-OH patients (33.3% in OH vs. 49.2% in non-OH, *p* = 0.19; 21.7% in OH vs. 26.5% in non-OH, *p* = 0.53) [19,25], in other two studies there were opposite results (42.86% in OH vs. 38.61% in non-OH, *p* = 0.61; 38% in OH vs. 20% in non-OH, *p* = 0.04) [20,21], whereas another one reported equal prevalence of HBP among the two groups (*p* = 1) [28], which makes it difficult to draw conclusions regarding the association between HBP and OH. In one study, a group of 26 patients with untreated HBP without PD served as control group for PD patients without HBP, revealing higher left ventricular mass (indexed to body surface) in both HBP and reverse dipper PD compared to non-reverse dipper PD patients [17].

OH was found in 941 patients (38.68% of the patients evaluated in this regard). In some studies, statistically significant associations were found between OH and clinical manifestations severity as well as abnormal imaging findings. OH was associated with more severe PD disease [16,19,20], higher rates of depression [16,34], anxiety [20] and fatigue [20,24], as well as more severe cognitive impairment [16,19,20,24,42], autonomic dysfunction [16,20] and motor symptoms [16,20], although not all these results were reproduced by other studies. A couple of them found no significant differences in disease severity [21,26,28,33,41] or motor symptoms [19,21,41] between OH and non-OH group. PD patients with OH also had smaller volumes of the caudate and right anterior ventral and left posterior putamen in one study [19]. Moreover, in de novo untreated PD, OH correlated with carotid artery thickening [37]. Interestingly, in one study dopaminergic treatment did not seem to increase the occurrence of OH [33].

SH was identified in 445 patients (27.76% of the patients evaluated in this regard). One study found that in patients with early PD and SH the motor symptoms and cognitive decline seemed to be worse compared to non-SH patients [42]. Moreover, SH was significantly correlated with restless legs syndrome [38], carotid artery thickening [37]/arterial stiffness [28] and an increased number of cerebral microbleeds in any brain region [25]. SH during sleep was shown to lead to NH, classified in 737 patients in our studies (38.91% of the patients investigated for NH). NH seemed to be more prevalent in PD patients experiencing restless legs syndrome [38] and advanced stages of PD [30], although these findings were not reproduced in all the studies—Oh et al. found no significant differences between PD patients with and without NH in terms of disease severity [40]. In other studies, NH was independently associated with arterial stiffness [28]/carotid artery thickening [37] and larger extent of white matter lesions in early PD [40].

Dipping status is detailed in Table 2 and Appendix A. One study did not mention the definitions used for the dipper profile [19]. The pathological nocturnal profiles are reverse dipping, reduced dipping, and extreme dipping, described in 477 (40.46% of the patients evaluated for this), 310 (35.67%) and 23 (5.1%) subjects, respectively. In one study, PD patients with reduced dipping were more prone to psychotic symptoms [35], but not to depression [34]. Those with reverse dipping had more severe non-motor manifestations such as cognitive decline [16,17,18,24], anxiety [16,17] and autonomic symptoms [16,17,23,39]. Even after adjusting for age, sex, disease duration, HBP, and intake of antihypertensives, this profile was strongly associated with cardiovascular dysautonomia, with a higher sensitivity and accuracy than OH [23]. Reverse dippers also had higher left ventricular mass and prevalence of left ventricular hypertrophy compared to PD patients without reverse dipping [17] and required higher doses of dopaminergic treatment [39]. Non-dippers were found to have more severe motor manifestations than dippers [43]. Conversely, one study found no significant differences between dipper, reduced dipper, and reverse dipper profiles in PD patients in terms of dopamine agonist usage, levodopa equivalent dose, disease duration [15] and stage or motor symptoms severity [15,41].

BP measurements are provided in Appendix A. Examples of devices used for ABPM are Mobil-Ograph NG, Spacelabs 90207, Fukuda Denshi, TM-2430, Norav Medical GmbH, and Boso GmbH. Daytime and nighttime were defined either according to the patient’s diary or a prespecified schedule. Measurements were taken variably (e.g., every 20 min during daytime + 30 min during nighttime, every 15 min during the whole day, every 30 min during daytime + 60 min during nighttime) and a variable number of readings was considered valid. Mean values for 24-h, diurnal and nocturnal SBP/DBP were recorded in most studies, with 6 studies not reporting any of these and 16 studies providing incomplete information. Higher mean values of 24-h and daytime SBP and DBP were significantly correlated with dementia [24] as well as deep or infratentorial cerebral microbleeds [25], whereas higher mean nocturnal SBP was associated with depression [36] and restless legs syndrome in PD patients [38].

Mean diurnal and nocturnal SBP/DBP load were reported only in 6 studies, whereas mean 24-h, diurnal, and nocturnal BP coefficient of variation were reported only in 7 studies. PD patients with higher BP fluctuations seemed to experience more commonly restless legs syndrome [38] and altered cognition in early PD stages [42]. Yoo et al. [19] found that increased BP variability is also associated with deep gray matter changes (inversely correlated with thalamus, globus pallidus and hippocampus volumes) in patients with early PD. Another interesting finding is that normotensive BP patients have higher BP variability than control subjects, which further highlights the role of dysautonomia in altering the circadian rhythms [27].

## 4. Discussion

### 4.1. Summary of Evidence

This systematic review offers an insight into the published studies that address BP patterns in patients with PD, as measured by ABPM. Many patients showed dysregulation of the BP circadian rhythm, as reflected by the increased prevalence of NH (38.91%) with reduced (35.67%) or reverse dipping (40.45%). In healthy controls and HBP patients, this pattern was found to be intimately associated with cardiovascular events and end-organ damage, predisposing to hypertensive cardiomyopathy, vascular thickening and cerebral small vessel disease [10]. Cognitive impairment and apathy are also common in this group of patients [10]. Considering the results previously presented, it seems legitimate to extrapolate at least some of these findings to the PD population. With emerging evidence that reverse dipping might be an early marker of dysautonomia in PD patients [22,44], we advocate for inquiring the dipper status in all patients with PD, even if they do not have OH or other autonomic manifestations.

Another finding of the studies included in this review is the high prevalence of OH, correlated with unfavorable outcomes such as more severe non-motor symptoms [16,19,20,24,34,42], atrophy of the striatum [19] and carotid artery thickening (the latter in the novo untreated PD patients) [37]. This prompts early detection and proper management of OH in PD patients. In this regard, one aspect needs clarification. The MDS Clinical Diagnostic Criteria for Parkinson’s Disease label OH occurring in the first 5 years of disease as a red flag for PD [52], but state that “this criterion is intended to identify the severe autonomic dysfunction associated with multiple system atrophy” and refers to higher orthostatic decrease of BP than that defining OH (≥30 mmHg in SBP *and*15 mmHg in DBP vs. ≥20 mmHg in SBP *or* ≥ 10 mmHg in DBP) [8,52]. Therefore, mild OH can occur in the early stages of PD [21]. It is of uttermost importance that the effects of coexistent OH and HBP in PD patients were not addressed in any study. Conflicting reports about the prevalence of HBP in PD patients with and without OH merely reflects the scarcity of information regarding their association.

With 27.76% of BP patients experiencing SH, SH also seems to be a common manifestation in this population. SH is relevant since it exacerbates pressure natriuresis during sleep, leading to sleep disturbances, nocturia and worsening of morning OH [9]. Dealing with the SH-OH overlap is a conundrum—apart from fact that the drugs used for OH induce or exacerbate SH and vice versa, some antihypertensives administered in the morning do not have an effect lasting until nighttime, leaving NH uncovered therapeutically. SH is not to be confused with primary HBP since their mechanisms differ substantially (dysautonomia vs. various mechanisms in primary arterial hypertension) and most of the patients with SH are normotensive while seated.

### 4.2. Limitations

The main limitation of this systematic review arises from the design of the studies included (i.e., cross-sectional) which does not allow us to establish causality between PD and the BP profiles. Another major limitation is the heterogeneity of the studies, both in terms of patient recruitment and manner of reporting the results. Notably, the inclusion and exclusion criteria for patient enrollment were significantly different among studies, with possible consequences on the BP patterns. For instance, HBP served as exclusion criterion in some studies, whereas in others it was not reported, consequently ignoring its interference with BP profiles. The intake of vasoactive drugs was also not consistently taken into consideration, with most of the studies not reporting it, withholding it or excluding the patients treated. Other potential confounders are the conditions predisposing to autonomic dysfunction (such as diabetes mellitus or other causes of peripheral neuropathy) or altered nocturnal BP profiles (such as REM sleep behavior disorder, obstructive sleep apnea, nocturia) which were not evaluated consistently. Mean BP load and coefficient of variation were reported only in 6 and 7 studies, respectively, making it difficult to make assumptions in this regard. Although of minor importance, the fact that the studies defined daytime and nighttime either according to the patient’s diary or a prespecified schedule might also interfere with the results provided that the patient is not asleep at the predicted nighttime.

## 5. Conclusions and Future Perspectives

Cardiovascular autonomic dysfunction is a common finding in patients with PD and has a negative impact on their quality of life, functionality and life expectancy, underpinning the need for proper assessment and management. ABPM is an accessible and feasible method for evaluating BP patterns. It can capture the extent of cardiovascular homeostasis disruption throughout a day in BP patients by offering insights into BP fluctuations and BP profiles regarding patient’s routine activities. It could also assist therapy based on real-life BP recordings. Although extensively validated for the assessment of HBP [8], ABPM has not yet been validated as a biomarker of PD-associated cardiovascular dysautonomia or a tool for assisting therapeutic interventions in this setting. An important drawback of this method is that it does not take into account the variability of BP patterns on a day-to-day basis [9], but this could possibly be overcome by repeated use or use over longer periods. Furthermore, we emphasize the need for thoroughly evaluating the interference of HBP in PD patients with dysautonomia and providing guideline therapeutic recommendations in case of HBP-OH overlap. In conclusion, we advocate for the validation and employment of ABPM in assessing and approaching cardiovascular autonomic dysfunction in PD patients with or without HBP, in an attempt to minimize its detrimental effects and to promote favorable functional outcomes and higher life expectancy in these patients.

## Figures and Tables

**Figure 1 jpm-11-00129-f001:**
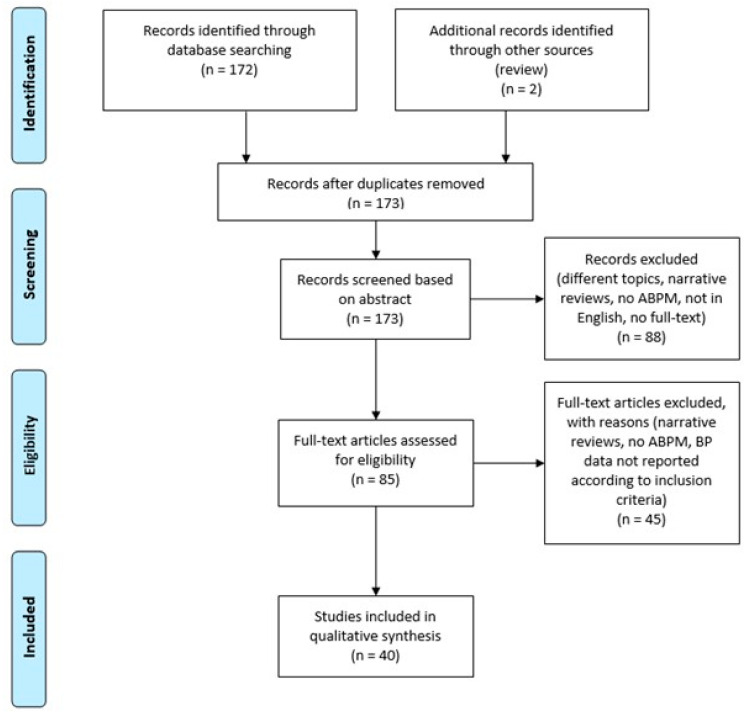
Flow diagram showing selection process.

**Table 1 jpm-11-00129-t001:** Demographic data and Parkinson’s Disease status.

No.	Study	No. of Patients	F/M (no.)	Mean Age (years)	Mean PD Duration (years)	Mean H&Y Stage	Mean Motor UPDRS	Mean Total UPDRS
1	Arici and Helvaci (2020) [15]	35	11/24	61.8 ± 9.1	10.3 ± 4.6	2.6	25 ± 8	-
2	Chen et al. (2020) [16]	103	43/58	66.6 ± 8.2	3	1.6	19.5 ± 14.9	37.1 ± 23.9
3	Di Stefano et al. (2020) [17]	52	16/36	68.7	6.7	-	-	-
4	Oka et al. (2020) [18]	75	48/27	72.2 ± 9.4	1.6 ± 1.6	-	20.3 ± 10.9	-
5	Yoo et al. (2020) [19]	98	44/53	69.4 ± 9.4	1	-	14.4 ± 7.9	21.6 ± 11.2
6	Kotagal et al. (2019) [12]	35	-	69.3	4.9	-	33.8	-
7	Li et al. (2019) [20]	150	72/78	64.7	-	-	-	-
8	Vallelonga et al. (2019) [21]	113	32/81	64.8 ± 10.2	6.5 ± 4.1	2	-	-
9	Vallelonga et al. (2019) [22]	72	18/54	70 ± 8	7.8 ± 6	-	-	-
10	Milazzo et al. (2018) [23]	114	35/79	64 ± 10	6 ± 4	-	-	-
11	Tanaka et al. (2018) [24]	137	74/53	64.1 ± 10.5	10.9 ± 6.2	3 ± 0.9	-	-
12	Yamashiro et al. (2018) [25]	128	72/56	64.4 ± 9.7	10.7 ± 6	2.9 ± 0.9	-	-
13	Franzen et al. (2017) [26]	27	13/14	67.5 ± 7.3	10.1 ± 6.1	2 ± 0.9	16.6 ± 7.2	-
14	Kanegusuku et al. (2017) [27]	21	10/11	66 ± 2	7.7 ± 0.5	2.6 ± 0.1	45 ± 3	-
15	Kim et al. (2017) [28]	125	68/57	68.5 ± 9.1	2.4 ± 3.5	1.6 ± 0.8	-	25.8 ± 19.7
16	Kim et al. (2017) [29]	99	50/49	68.9 ± 10.1	1.6 ± 1.7	1.5 ± 0.7	13.8 ± 10.3	22.1
17	Vetrano et al. (2017) [30]	167	59/108	73.4 ± 7.6	-	-	24	-
18	Vichayanrat et al. (2017) [31]	51	21/30	67	-	-	-	-
19	Kang et al. (2016) [32]	46	19/27	66.9 ± 9.1	2.8 ± 3.1	2.3 ± 1	27.1 ± 15.4	-
20	Kim et al. (2016) [33]	188	101/87	68.4 ± 10.4	3.4 ± 3.7	1.8 ± 0.9	-	27.2 ± 25.2
21	Park et al. (2016) [34]	129	79/50	68.1 ± 10.4	2 ± 1	1.8 ± 0.7	14.5 ± 10.4	24.6 ± 17.5
22	Stuebner et al. (2015) [35]	21	6/15	63.9 ± 7.6	6.6 ± 7	1.7	-	28 ± 13
23	Vetrano et al. (2015) [36]	125	40/85	72.7 ± 7.8	-	-	-	40.3
24	Fanciulli et al. (2014) [14]	16	5/11	-	4	-	-	-
25	Kim et al. (2014) [37]	65	41/24	67.7 ± 11	2.1 ± 1	1.6 ± 0.7	-	24.4 ± 15.4
26	Oh et al. (2014) [38]	225	131/94	68 ± 10.9	1.8 ± 0.8	1.7 ± 0.8	-	25.2 ± 18.5
27	Pilleri et al. (2014) [13]	61	-	65.6 ± 9.3	11.7 ± 4.6	2.8 ± 0.5	36.2 ± 14.6	-
28	Berganzo et al. (2013) [39]	111	49/62	67.9 ± 8.6	6.6 ± 5.2	2.3	29.1	-
29	Oh et al. (2013) [40]	129	81/48	68.9 ± 10.1	2.1 ± 1	1.8 ± 0.7	15 ± 10.9	25.3 ± 18.4
30	Oh et al. (2013) [41]	69	41/28	68.7 ± 9.3	1.7 ± 0.9	1.8 ± 0.8	-	26.6 ± 19.7
31	Kim et al. (2012) [42]	87	52/35	67.5 ± 9.2	1.8 ± 0.8	1.7 ± 0.7	-	22.4 ± 16.6
32	Manabe et al. (2011) [43]	37	22/15	71.8	3.4	2.7	26.5	-
33	Sommer et al. (2011) [44]	40	20/20	69.9	4.1	-	-	-
34	Reimann et al. (2010) [45]	26	7/19	64.5 ± 8.6	-	-	-	-
35	Schmidt et al. (2009) [46]	23	6/17	65 ± 8	-	-	-	-
36	Ejaz et al. (2006) [47]	13	4/9	76.5 ± 5.8	-	-	-	-
37	Sigurdardóttir et al. (2001) [48]	10	8/2	64.1 ± 8.5	18.4 ± 5	-	-	-
38	Plaschke et al. (1998) [49]	24	11/13	67.1	9.3	3.4	-	-
39	Senard et al. (1992) [50]	38	20/18	66.5	9	3.6	-	-
40	Micieli et al. (1989) [51]	5	2/3	66.2 ± 7.4	1.1 ± 0.4	1.6 ± 0.9	-	-
TOTAL		3090	1431/1550 (48%/52%)					

F = female, M = male, H&Y = Hoehn and Yahr, PD = Parkinson’s disease, UPDRS = Unified Parkinson’s Disease Rating Scale.

**Table 2 jpm-11-00129-t002:** Blood Pressure patterns in Parkinson’s Disease patients.

Blood Pressure Patterns	Yes No. of Patients (%)	No No. of Patients (%)	Total No. of Patients	No. of Studies
High blood pressure	938 (38.13%)	1522 (61.87%)	2460	26
Supine hypertension	445 (27.76%)	1158 (72.24%)	1603	15
Orthostatic hypotension	941 (38.68%)	1492 (61.32%)	2433	30
Nocturnal hypertension	737 (38.91%)	1157 (61.09%)	1894	22
Reverse dipping (<0%)	477 (40.46%)	702 (59.54%)	1179	18
Reduced dipping (0–10%)	310 (35.67%)	559 (64.33%)	869	14
Normal dipping (10–20%)	106 (23.5%)	345 (76.5%)	451	5
Extreme dipping (>20%)	23 (5.1%)	428 (94.9%)	451	5
Non-dipping (<10%)	1744 (75.53%)	565 (24.47%)	2309	27
Dipping (>10%)	565 (24.47%)	1744 (75.53%)	2309	27
Reduced dipping and dipping (>0%)	702 (59.54%)	477 (40.46%)	1179	18

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
