# Peer review of "Blood Pressure Patterns in Patients with Parkinson’s Disease: A Systematic Review"

_jpm, 2021, doi:10.3390/jpm11020129_

Round 1
Reviewer 1 Report
This is a very interesting review on an interesting topic, further characterizing the profile of blood pressure throughout the day in Parkinson’s disease patients. The authors convincingly argue for the importance of the topic, particularly given the known phenomenon of autonomic dysregulation in PD, and the likely need for understanding the effects in the PD context, as well as the possibility of identifying distinct subtypes within PD that may feature different BP-related symptom constellations. The review is largely well-written, but a few modifications could improve it.
Many associations are reported, but it is difficult to differentiate between when a single study examined an association and found one, vs if multiple studies looked, but no association was found in some. There seem to be very few instances of conflicting evidence reported, and it isn’t clear to me if that is because there has been very little overlap in the measures examined between studies.
Essentially no comment on the various control groups in the studies, and how they compared to PD. Probably worth a section or a comment in subsections where it applies
Given the introductory commentary about the potential difficulty specifically where OH and HBP coexist, it is surprising that this overlap does not appear to be addressed in the results? If no study examined this, that is also a result that should definitely be mentioned.
In the discussion it is stated that “Most of the patients showed dysregulation of the BP circadian rhythm, as reflected by the increased prevalence of NH (38.91%) with reduced (35.67%) or reverse dipping (40.45%)”. It is not clear to me how ~40% reflects “most” patients; perhaps the wording is not quite what the authors intended to say.
Some visual representation of the findings would be useful. The tables are informative, but also difficult to compare between studies and subgroups over multiple pages.
Author Response
Reviewer 1:
This is a very interesting review on an interesting topic, further characterizing the profile of blood pressure throughout the day in Parkinson’s disease patients. The authors convincingly argue for the importance of the topic, particularly given the known phenomenon of autonomic dysregulation in PD, and the likely need for understanding the effects in the PD context, as well as the possibility of identifying distinct subtypes within PD that may feature different BP-related symptom constellations. The review is largely well-written, but a few modifications could improve it.
We thank the reviewer for reading through and summarizing our review. The suggestions helped us to improve our work.
Many associations are reported, but it is difficult to differentiate between when a single study examined an association and found one, vs if multiple studies looked, but no association was found in some. There seem to be very few instances of conflicting evidence reported, and it isn’t clear to me if that is because there has been very little overlap in the measures examined between studies.
In the revised form of the manuscript, we made some changes in order to clarify the fact that only some studies performed subanalysis of BP profiles, therefore the correlations found virtually apply to a small number of patients. Moreover, we added info about conflicting results. (please see Page 6, lines 193-195, lines 198-200; Page 7, lines 222-225, lines 238-240, lines 256-260).
Essentially no comment on the various control groups in the studies, and how they compared to PD. Probably worth a section or a comment in subsections where it applies.
We mentioned the various control groups in the studies (Page 5, lines 179-184), but we considered that comparing them to PD does not fall within the scope of this systematic review, especially since no subanalysis of BP profiles was performed within these groups.
Given the introductory commentary about the potential difficulty specifically where OH and HBP coexist, it is surprising that this overlap does not appear to be addressed in the results? If no study examined this, that is also a result that should definitely be mentioned.
Indeed, no study considered the effects of coexistent OH and HBP- in the revised form of the manuscript, we clearly stated this (Page 6, lines 204-207; Page 7, lines 208-212; Page 9, lines 308-311).
In the discussion it is stated that “Most of the patients showed dysregulation of the BP circadian rhythm, as reflected by the increased prevalence of NH (38.91%) with reduced (35.67%) or reverse dipping (40.45%)”. It is not clear to me how ~40% reflects “most” patients; perhaps the wording is not quite what the authors intended to say.
We replaced the word “most” by “many” (Page 8, line 287).
Some visual representation of the findings would be useful. The tables are informative, but also difficult to compare between studies and subgroups over multiple pages.
We couldn’t figure out a visual representation for BP profiles since we reported them only as frequencies and couldn’t draw conclusions regarding their interrelationship. However, in the revised version of the manuscript, we did draw a table with the findings that might help the reader to visualize them more easily (Page 8, line 262).
Reviewer 2 Report
Report on manuscript
Abstract
The authors systematically reviewed papers describing blood pressure patterns found in patients with Parkinson’s disease, as measured by 24-hour ambulatory blood pressure monitoring. The blood pressure patterns included hypertension, orthostatic hypotension, supine hypertension, nocturnal hypertension and dipping status. These patterns were associated with cardiovascular autonomic impairment. Most of the patients showed dysregulation of the BP circadian rhythm, but the design of the studies included in the review cannot establish causality between PD and the BP profiles.
The systematic review seems to be comprehensive and properly described, including the limitations of the study. We only have comments on the Tables 3 and 4.
- In Table 3, table cells without information are denoted by a hyphen, while in Table 4, by NA.
- In Table 4 there is no final row with concluding information and remarks. The table only presents the hypertension information that appears in each paper.
Author Response
Reviewer 2:
The authors systematically reviewed papers describing blood pressure patterns found in patients with Parkinson’s disease, as measured by 24-hour ambulatory blood pressure monitoring. The blood pressure patterns included hypertension, orthostatic hypotension, supine hypertension, nocturnal hypertension and dipping status. These patterns were associated with cardiovascular autonomic impairment. Most of the patients showed dysregulation of the BP circadian rhythm, but the design of the studies included in the review cannot establish causality between PD and the BP profiles.
We thank the reviewer for reading through and summarizing our review.
The systematic review seems to be comprehensive and properly described, including the limitations of the study. We only have comments on the Tables 3 and 4.
We thank the reviewer for the remarks.
In Table 3, table cells without information are denoted by a hyphen, while in Table 4, by NA.
In the revised form of the manuscript, we used a hyphen in all the tables, as you suggested.
In Table 4 there is no final row with concluding information and remarks. The table only presents the hypertension information that appears in each paper.
We did not attach a final row with concluding information and remarks because it was impossible to calculate the total mean values of 24-h, diurnal and nocturnal SBP/DBP since they were reported as mean values ± standard deviation (perhaps this statistical analysis could have been employed in a meta-analysis).
Reviewer 3 Report
Overall I found the paper really well-written and organized. The results are presented clearly, and the limitation of the study are carefully analyzed and presented.
All in all, I think the authors do a great work on the manuscript. It just needs minor correction - please, present Tables 1-3 as Supplementary Materials. It would improve readability.
Author Response
Reviewer 3:
Overall I found the paper really well-written and organized. The results are presented clearly, and the limitation of the study are carefully analyzed and presented.
We thank the reviewer for the remarks.
All in all, I think the authors do a great work on the manuscript. It just needs minor correction - please, present Tables 1-3 as Supplementary Materials. It would improve readability.
In the revised form of the manuscript, we presented all the tables as Supplementary Materials, as you suggested (except for Table 1 which contains demographic data that are probably worth being seen in the main text).